

# Digital competence of faculty members in health sciences measured *via* self-reflection: current status and contextual aspects

Halil Ersoy[1], Cigdem Baskici[2], Aydan Aytar[3], Raimonds Strods[4], Nora Jansone Ratinika[4], António Manuel Lopes Fernandes[5], Hugo Neves[5], Aurelija Blaževičienė[6], Alina Vaškelytė[6], Camilla Wikström-Grotell[7], Heikki Paakkonen[8], Anne Söderlund[9], Johanna Fritz[9] and Sultan Kav[10]

[1] Computer Education and Instructional Technology, Başkent University, Ankara, Türkiye
[2] Department of Healthcare Management, Başkent University, Ankara, Türkiye
[3] Gulhane Faculty of Physiotherapy and Rehabilitation, University of Health Sciences, Ankara, Türkiye
[4] Centre for Educational Growth, Riga Stradins University, Riga, Latvia
[5] Nursing School of Coimbra, Escola Superior de Enfermagem de Coimbra, Coimbra, Portugal
[6] Department of Nursing, Lithuanian University of Health Sciences, Kaunas, Lithuania
[7] Graduate School and Research, Arcada University of Applied Sciences, Helsinki, Finland
[8] Advanced Clinical Care, Arcada University of Applied Sciences, Helsinki, Finland
[9] School of Health Care and Social Welfare, Mälardalen University, Västrås, Sweden
[10] Department of Nursing, Başkent University, Ankara, Türkiye

Corresponding author
Halil Ersoy, halilersoy@gmail.com

## ABSTRACT

This descriptive study aims to determine the digital competence level of faculty members who teach in the health sciences, empirically considering possible contextual aspects. Two data collection instruments were used: a self-reflection questionnaire to assess digital competence, and a survey querying demographics and aspects of teaching and learning context. In total, 306 health sciences faculty members from six universities voluntarily participated the study. The results revealed that a majority of the faculty members have intermediate (integrator or expert) level of digital competence, which is described as being aware of the potential use of digital technology in teaching and having a personal repertoire for its use under various circumstances. Age, digital teaching experience, perception of work environment, and previous teaching experience in fully- or partially-online courses were identified as influencing factors for digital competence. Faculty members in health sciences were able to integrate digital technologies in their teaching practices. Health education institutions may facilitate the use of digital technologies in teaching and learning environments. Moreover, institutions or stakeholders should consider that digital competence requires practice and experience in meaningfully-designed digital environments and tools.

## INTRODUCTION

As in all fields of education, the use of digital technologies (DT) might facilitate the creation of learning environments that are as effective as face-to-face instruction in the health sciences (*Car et al., 2019*; *Kyaw et al., 2019*). The efficacy of online or e-learning has been demonstrated in the literature when the digital tools are properly designed and administrators and students can meet adequate digital competency (DC) levels (*Noesgaard & Ørngreen, 2015*; *Thalheimer, 2017*). As the use and impact of digital innovations and interventions in health education grow, it is critical to improve the DC of faculty members in these fields (*Basilotta et al., 2022*; *Cabero-Almenara et al., 2021a*; *Choules, 2007*). Enhancing the DC of faculty members should be a key focus for institutions, and may help meet the rising demand for higher education as well as the changing needs and expectations of today's students (*Spante et al., 2018*).

In their systematic review, *Batanero et al. (2020)* found that the majority of studies demonstrated a lack of information and communication technologies (ICT) skills and regarding training for academicians. Similarly, *Akram et al. (2021)* used a technological pedagogical content knowledge (TPACK) model to assess the technology, pedagogy, and content knowledge of faculty and found that while other two sub-domain knowledge levels were adequate, the technological domain was low.

*From (2017)* called the digital skills of faculty members a "pedagogical digital competence" (PDC). These skills are defined as "teachers' ability to use ICT in their teaching practice". *Guillén-Gamez et al. (2021)* added that PDC includes skills as well as the attitude to improve in the personal and academic use of ICT in learning.

The broader definitions of DC were constructed in more comprehensive frameworks because the competency of a faculty member may be measured in multiple but interconnected sub-skills in task-related sub-domains of their profession (*International Society for Technology in Education Standards (ISTE), 2021*; *UNESCO, 2011*). Consequently, assessing DC for an academician is a challenging process. For this purpose, the DigComEdu framework (*DigCompEdu, 2021*) was constituted by the European Research Center in 2017 with both a definition and a self-reported assessment tool for DC (*Redecker, 2017*). According to the DigCompEdu framework, DC is:

> "… an ability to use digital technologies not only to enhance teaching, but also for their [educators'] professional interactions with colleagues, learners, parents and other interested parties, for their individual professional development and for the collective good and continuous innovation in the organization and the teaching profession" (*Redecker, 2017*, p. 19).

In the DigCompEdu framework, the DC of educators was defined in six areas: professional engagement, digital resources, teaching and learning, assessment, empowering learners, and facilitating learners' DC (*Redecker, 2017*). The framework also provided a DC assessment tool in a self-reflection questionnaire to portray the DC of faculty members in those six areas. More details about that tool are given in the Data Collection section.

It is important to determine the DC level of faculty members, yet this alone is not robust enough for the successful design and implementation of proper DC development. Understanding the related factors influencing DC levels can help in providing effective and tailored support mechanisms. In a number of studies, variables such as gender, age, generation, academic level (*Guillén-Gamez et al., 2021*; *Jansone-Ratinika et al., 2021*; *Jorge-Vazquez et al., 2021*), teaching experience (*Batanero et al., 2021*; *Cabero-Almenara et al., 2021b*), digital technology experience in teaching (*Ghomi & Redecker, 2019*; *Hatlevik, 2017*; *Lucas, Dorotea & Piedade, 2021*; *Tondeur et al., 2018*), and availability and accessibility in technological infrastructure with support (*Cattaneo, Antonietti & Rauseo, 2022*; *Mohan et al., 2020*) are found to be related in DC.

Moreover, the experiences during compulsory distance education during the COVID-19 pandemic could be effectual in evaluating the DC of faculty members. During that time teachers were forced to utilize many new technological tools and pedagogical strategies imposed by not only their institutions but also the facilities that were available at their workplace and home (*Gonzalez, Ponce & Fernandez, 2023*). They may have had to gain new DCs about using ICT to survive in teaching.

To improve the digital competence of faculty members in health sciences in higher education, an Erasmus+ Project, called DITEPRACT (Digital and Hybrid Teaching and Learning of Practical Skills in Higher Education), was initiated by six partner universities from Finland, Latvia, Lithuania, Portugal, Sweden, and Türkiye in 2021 (*Jansone-Ratinika, Surakka & Wikstrom-Grötell, 2023*). The objective of the project was to discover the best DT experiences or tools utilized, especially in subject-specific practical teaching in the disciplines. DCs are gained from DT, therefore, it was important to make a valid assessment of DC levels of faculty members with related contextual aspects.

Accordingly, this study aims to assess the DC of faculty members teaching practical skills in the health sciences and to understand the contextual aspects of their competencies. The two research questions were as followings:

1. What is the DC level of the faculty members?

2. Do the demographic characteristics and teaching experience of the faculty members affect their DC level?

## MATERIALS AND METHODS

The primary aim of this descriptive study is "to provide accurate description of the status or characteristics of a situation or phenomenon" (*Johnson & Christensen, 2004*, p. 347). A two-part survey was administered to faculty members from six universities in the DITEPRACT project to answer our research questions.

### Participants

The participants were faculty members from six partner universities who taught practical skills during the 2020–2021 academic year. A total of 306 faculty members from diverse departments in health sciences participated in the study after being selected using a convenience sampling strategy. The average age was 46.7, and the average teaching experience was 14.6 years (Fig. 1).

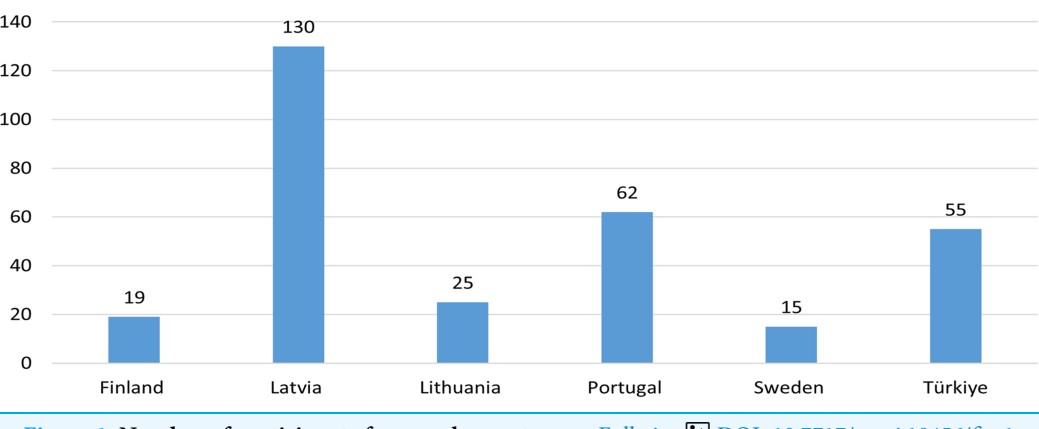

Figure 1 **Number of participants from each country.**

## Data collection instruments and process

For the first research question, a "DigCompEdu CheckIn Self-Reflection Tool"
questionnaire (*DigCompEdu, 2021*) was used to assess the DC of faculty members. The
questionnaire was first published in English in 2017 by the "European Commission's Joint
Research Center" as "European Framework for the Digital Competence of Educators:
DigCompEdu" (*DigCompEdu, 2021*). The significance of the DigCompEdu is that "the
focus is not on technical skills, rather, the framework aims to detail how digital
technologies can be used to enhance and innovate education and training." (*DigCompEdu,
2021.*, p.1). The questionnaire includes 22 competency statements in six domains:
"professional engagement, digital resources, teaching and learning, assessment,
empowering learners, and facilitating learners' DC" (*DigCompEdu, 2021*). There are five
options in each question scored from 0 to 4 points. The higher score in a question
means higher competence described in the question. The maximum score is 88 when a
participant marks highest-competency statement in each question. The total score of a
participant indicates his or her DC level according to predefined six competence levels, as
explained in Fig. 2.

At the beginning of the questionnaire, the participants were given a brief description of
competency levels from A1 to C2 and then asked to rate their DC level from this range
(pre-self-assessed DC score). Next, they responded to all items and were asked again to rate
their DC from the same range (post-self-assessment score). Once the participants
completed the evaluation tool, the researchers calculated the DC points of faculty members
based on their responses (scale-based DC score). According to the DigCompEdu
Framework (*DigCompEdu, 2021*), each participant was assigned to one of the following
competency categories: newcomer (A1), explorer (A2), integrator (B1), expert (B2), leader
(C1), or pioneer (C2).

At the beginning of the study, written permission was obtained *via* email from the
DigCompEdu research team. The original questionnaire was in English, with additional
versions available in some European languages (*DigCompEdu, 2021*). Two partners used
the English version and others preferred a version in their local language. The Cronbach's
α coefficient of questionnaire was calculated as 0.915 for the study.

| A1 | **Newcomer** (0-19)<br>Aware and uses DT, but without consistency. Needs encouragement. |
| A2 | **Explorer** (20-32)<br>Aware and uses DT, but without consistency. Needs encouragement. |
| B1 | **Integrator** (33–49)<br>Using DT in education consistently and integrates in various practices, but needs time and experience to make right decisions about DT. |
| B2 | **Expert** (50-65)<br>Uses a variety of DT consistently, creatively, and critically. Open for exploration. |
| C1 | **Leader** (66-80)<br>Has a variety of DT in their repertoire. Guides and leads peers regarding DT usage. |
| C2 | **Pioneer** (81-88)<br>Uses DT and experiments with it to develop new pedagogical practices. |

**Figure 2** DigCompEdu levels, role describers (*DigCompEdu, 2021*), range of scores (*Toker et al., 2021*), and explanations.

For the second research question, a follow-up was conducted to inquire about the demographic and descriptive characteristics of the participants, including academic title, age, duration of teaching experience, time spent using DT, and digital tools used for learning. Likert questions about personal use patterns of DT and about the institutional facilities were also used in this section as described in "DigComEdu CheckIn Self Reflection Tool" (*DigCompEdu, 2021*). Data were collected from participants online using Google® Forms. Participation was voluntary.

## Data analysis

After data collection, the data from six universities were accumulated into a single dataset. For categorical variables, like demographics and having available DT at work or at home, frequency distributions were used; for numerical variables like DC scores, descriptive statistics (means and standard deviations) were calculated. Multiple linear regression analysis was used to determine the influencing factors such as age, teaching experience, perception of the work environment, and having experience teaching a fully- or partially- (in blended or hybrid form) online course.

A paired t-test was used to determine the difference between perceived (pre- and post-) and real (scale-based) levels of DC according to the self-assessment questionnaire. Statistical analyses were carried out using the SPSS (v. 26) software at a significance level of 5% ($\alpha = 0.05$).

To answer the second research question, the following nine hypotheses (Fig. 3), were tested:

To test the first six hypotheses ($H_1$, $H_2$, $H_3$, $H_4$, $H_5$, $H_6$), an explanatory multiple linear regression model was proposed, as illustrated in Eq. 1, below.

$$DC = \beta_0 + \beta_1 Age + \beta_2 TE + \beta_3 DTE + \beta_4 PWE + \beta_5 FTEBC + \beta_6 PTEBC + \varepsilon_i \qquad (1)$$

*Hypothesis 1 (H₁):* Age has a significant and negative effect on DC.

*Hypothesis 2 (H₂):* Teaching experience (TE) has a significant and positive effect on DC.

*Hypothesis 3 (H₃):* Using DT in teaching (DTE) has a significant and positive effect on DC.

*Hypothesis 4 (H₄):* Perception of work environment (PWE) has a significant and positive effect on DC.

*Hypothesis 5 (H₅):* Having experience in teaching a fully online course before COVID-19 (FTEBC) has a significant and positive effect on DC.

*Hypothesis 6 (H₆):* Having experience in teaching a partially online course (in blended or hybrid form) before COVID-19 (PTEBC) has a significant and positive effect on DC.

*Hypothesis 7 (H₇):* DC has a significant difference between Pre-self-assessment score (PRES) and Post-self-assessment score (POSTS).

*Hypothesis 8 (H₈):* DC has a significant difference between PRES and SBS.

*Hypothesis 9 (H₉):* DC has a significant difference between POSTS and SBS.

**Figure 3 Nine hypothesis about contextual aspect related to DC.**

**$DC_i$:**     *Digital competency score of faculty members*

**$Age_i$:**     *Age of faculty members in years*

**$TE_i$:**     *Teaching experience in higher education of faculty members in years*

**$DTE_i$:**     *Using DT in teaching of faculty members in years*

**$PWE_i$:**     *The mean of perception of the work environment of faculty members*

**$FTEBC_i$:**     *Experience in teaching (in years) a fully online course (in blended or hybrid form) before COVID-19*

**$PTEBC_i$:**     *Experience in teaching (in years) a partially online course (in blended or hybrid form) before COVID-19*

**$\varepsilon_i$:**     *The error (residual) term in the regression model*

To test the remaining hypotheses ($H_7$, $H_8$, $H_9$), paired sample t-tests were employed.

## Ethical considerations

Ethical approval was granted by the ethics board committee of the Başkent University Academic Evaluation and Assessment Coordination Office on May 26th, 2021, with document number "E-62310886-604.02.01-35305". The prospective participants were informed about their rights to decline the involvement to the study and to withdraw from the data collection at any time. The digital informed consent from participants was received at the beginning of the online survey by providing the aims and scope of data collection.

Permission to use the DigComEdu CheckIn Self Reflection Tool was obtained by email from the DigComEdu research team.

## RESULTS

Table 1 shows the descriptive characteristics of the participants. The average age was 46.7 ± 11.7 years, with 72.5% of participants falling between the ages of 25 and 55. The average teaching experience of participants in higher education was 14.6 ± 10.3 years. The

**Table 1 The distribution of participants by their descriptive characteristics.**

|  | *n* | % |
|---|---|---|
| **Age** | | |
| 25–55 | 222 | 72.5 |
| 56–78 | 84 | 27.5 |
| **Gender** | | |
| Female | 246 | 80.4 |
| Male | 57 | 18.6 |
| Prefer not to say | 3 | 1.0 |
| **Academic title** | | |
| Prof. | 81 | 26.5 |
| Assoc. Prof. | 33 | 10.8 |
| Assist. Prof. | 73 | 23.9 |
| Research assistant | 52 | 17.0 |
| Lecturer | 67 | 21.9 |
| **Years of teaching experience in higher education** | | |
| 1–20 | 223 | 72.9 |
| 21–56 | 83 | 27.1 |
| **Years of DT usage experience in teaching** | | |
| 1–20 | 280 | 91.5 |
| 21–40 | 26 | 8.5 |
| **Main profile of the students*** | | |
| Undergraduate | 260 | 85 |
| Graduate master | 152 | 49.7 |
| Graduate doctorate | 48 | 15.7 |
| Adult students full-time | 24 | 7.8 |
| Adult students part-time | 18 | 5.9 |

**Note:**
  * More than one option could be marked.

average time spent using DT in teaching was 10.7 ± 7.6 years. For 85% of the participants, the main profile of the students was undergraduate.

Table 2 summarizes the availability of regular or sophisticated DT for teaching for the participants. The majority of participants stated that they used presentations (98%) and online communication tools (96.4%) for teaching. The least used digital tools were wikis and blogs (9.5%). In terms of the university's provision of DT for teaching/learning, 96.1% of the participants had synchronous communication tools and 90.2% had asynchronous communication tools. More than half of them (55.9%) stated that they had sophisticated or field-specific software. The least owned advanced tools are augmented/virtual reality tools (24.8%).

Table 3 summarizes the participants' private use of DT as well as the criteria met by their work environment. A total of 86.6% of participants stated that they used the Internet extensively and competently, and 88.3% found working with computers and other technical equipment to be simple. In the fourth statement, 62.4% of participants said they

**Table 2 Distribution of participants according to digital technologies used for teaching.**

| | No | | Yes | |
|---|---|---|---|---|
| | *n* | % | *n* | % |
| **Digital tools already used for teaching and learning**[§] | | | | |
| Presentations | 6 | 2.0 | 300 | 98.0 |
| Digital posters, mind maps, planning tools | 173 | 56.5 | 133 | 43.5 |
| Watching videos/listening to audios | 23 | 7.5 | 283 | 92.5 |
| Digital quizzes or polls (*i.e.*, Kahoot, Mentimeter, *etc.*) | 136 | 44.4 | 170 | 55.6 |
| Blogs or wikis | 277 | 90.5 | 29 | 9.5 |
| Creating videos/audios | 154 | 50.3 | 152 | 49.7 |
| Online/virtual learning environments (Moodle, *etc.*) | 74 | 24.2 | 232 | 75.8 |
| Online communication tools (Zoom, MS Teams, Skype, Google Meet, *etc.*) | 11 | 3.6 | 295 | 96.4 |
| Using any digital tools in class | 2 | 0.7 | 304 | 99.3 |
| **Digital technologies for teaching/learning provided by the university**[§] | | | | |
| Learning management system or virtual learning environment (like Moodle/Blackboard- *etc.*) | 60 | 19.6 | 246 | 80.4 |
| Student enrolment and grading system (course lists, grading, attendance, *etc.*) | 60 | 19.6 | 246 | 80.4 |
| Asynchronous communication tools (email, SMS, messaging) | 30 | 9.8 | 276 | 90.2 |
| Synchronous communication tools (Zoom, MS Teams, Skype *etc.*) | 12 | 3.9 | 294 | 96.1 |
| Online or electronic measurement and evaluation systems | 127 | 41.5 | 179 | 58.5 |
| **Having sophisticated or field-specific DT for teaching/learning**[§] | | | | |
| Augmented/virtual reality tools or environments | 230 | 75.2 | 76 | 24.8 |
| Simulations | 179 | 58.5 | 127 | 41.5 |
| Software or applications for specific tasks (*e.g.*, for advanced calculation, data analysis, *etc.*) | 135 | 44.1 | 171 | 55.9 |
| Special institutional membership for certain online collections/databases | 141 | 46.1 | 165 | 53.9 |
| Other | 293 | 95.8 | 13 | 4.2 |
| Having any of them | 9 | 2.9 | 297 | 97.1 |

**Note:**
[§] More than one option could be marked.

**Table 3 Distribution of participants by their use of DT at private life and at work environment.**

| | Disagree | | Neither agree nor disagree | | Agree | |
|---|---|---|---|---|---|---|
| | *n* | % | *n* | % | *n* | % |
| **How would you describe yourself and your private use of digital technologies?** | | | | | | |
| I find it easy to work with computers and other technical equipment | 5 | 1.6 | 36 | 11.8 | 265 | 86.6 |
| I use the Internet extensively and competently | 3 | 1 | 33 | 10.8 | 270 | 88.3 |
| I am open and curious about new apps, programs, resources | 7 | 2.3 | 50 | 16.3 | 249 | 81.4 |
| I am a member of various social networks | 54 | 17.6 | 61 | 19.9 | 191 | 62.4 |

|  | Disagree | | Neither agree nor disagree | | Agree | |
| --- | --- | --- | --- | --- | --- | --- |
|  | n | % | n | % | n | % |
| **How well does your work environment meet the following criteria?** | | | | | | |
| The department invests in updating and improving the technical infrastructure | 13 | 4.2 | 41 | 13.4 | 252 | 82.3 |
| The department provides the necessary technical support | 25 | 8.2 | 48 | 15.7 | 233 | 76.1 |
| Students have access to digital devices | 12 | 3.9 | 43 | 14.1 | 251 | 82 |
| The internet connection of the department is reliable and fast | 28 | 9.2 | 52 | 17 | 226 | 73.9 |
| The department supports the development of my digital competence, *e.g.*, through continuous professional development activities | 20 | 6.5 | 53 | 17.3 | 233 | 76.2 |

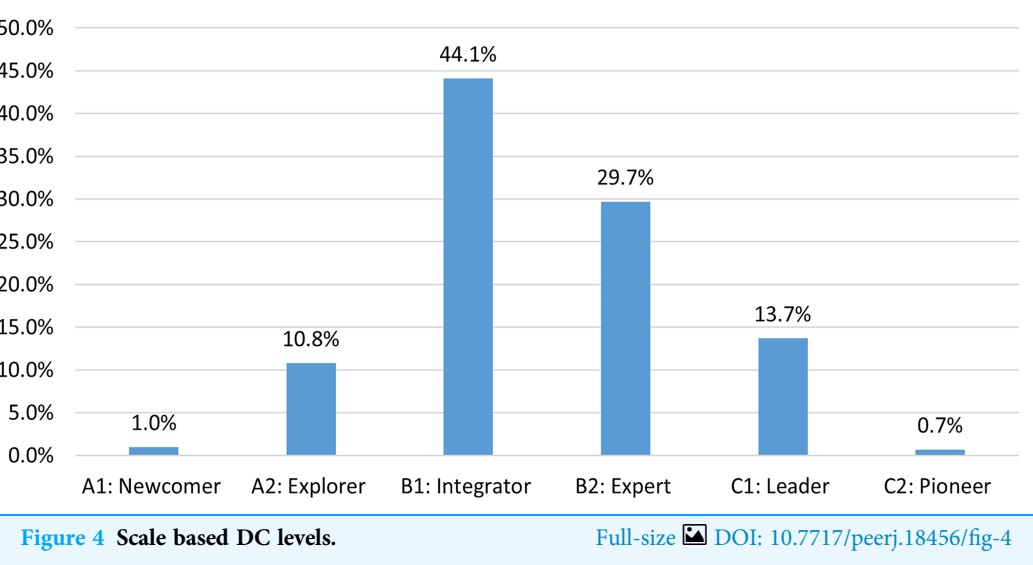

**Figure 4 Scale based DC levels.**

subscribed to various social networks. Considering the participants' assessments of the work environment, 82.3% believed that their department invests in updating and improving the technical infrastructure. Moreover, 73.9% of the participants found the department's internet connection to be reliable and fast in all criteria evaluations.

Figure 4 summarizes the DC of faculty members based on scale-based scores (SBC). In total, the integrator level (B1, 44.1%) had the largest proportion following the expert level (B2, 29.7%). The lowest (A1) and highest proficiency (C2) categories had the smallest number of participants, at 1% and 0.7%, respectively.

In Table 4, it is shown that the regression model was overall significant at $\alpha = 0.05$ level and the coefficients of age, DTE, PWE, FTEBC, and PTEBC were significant (95% confidence interval), except for TE. The signs of the regression models' coefficients should be checked to thoroughly examine the directions of the impacts of independent variables

**Table 4 The outline of multiple linear regression analysis[a,d,e].**

| R² | Adjusted R² | SE | F | p | Independent variable | Unstandardized coefficients | | t | p | Collinearity statistics | | Accepted/ Rejected |
|---|---|---|---|---|---|---|---|---|---|---|---|---|
| | | | | | | βi | SE | | | Tol. | VIF | |
| 0.164 | 0.147 | 12.069 | 9.672 | 0.000[b] | Constant | 44.806 | 5.161 | 8.682 | 0.000[b] | | | |
| | | | | | Age | −0.428 | 0.095 | −4.511 | 0.000[b] | 0.388[c] | 2.580[c] | $H_1$ accepted |
| | | | | | TE | 0.186 | 0.111 | 1.666 | 0.097 | 0.366[c] | 2.731[c] | $H_2$ rejected |
| | | | | | DTE | 0.368 | 0.109 | 3.374 | 0.001[b] | 0.704[c] | 1.419[c] | $H_3$ accepted |
| | | | | | PWE | 0.787 | 0.198 | 3.969 | 0.000[b] | 0.979[c] | 1.022[c] | $H_4$ accepted |
| | | | | | FTEBC | 5.491 | 2.140 | 2.566 | 0.011[b] | 0.979[c] | 1.021[c] | $H_5$ accepted |
| | | | | | PTEBC | 7.209 | 2.220 | 3.247 | 0.001[b] | 0.932[c] | 1.073[c] | $H_6$ accepted |

Notes:
[a] Dependent Variable: DC
[b] The regression model and its coefficients are statistically significant at α = 0.05 level.
[c] Due to the values of Tolerance (Tol.) and VIF are not lower/higher than cutoff values (Tol. (0.1) and VIF (10)), it can be expressed that there is no multicollinearity in the model (*Hair et al., 2009*).
[d] According to the results of Jarque-Bera Normality Test, it can be articulated that the data of the variable is distributed normally at the significance level of α = 0.05 level.
[e] According to the result of White Homoscedasticity Test, it can be stated that there is no heteroscedasticity in the model significance level of α = 0.05 level.

**Table 5 Summary of the paired sample t-test.**

| Dependent variable | Pair ID | | t | p | Result | Accepted/ Rejected |
|---|---|---|---|---|---|---|
| DC score | Pair 1 | Pre and post self-assessment | −4.81 | 0.000* | $\bar{X}_{\text{pre self-assessment}} < \bar{X}_{\text{post self-assessment}}$ | $H_7$ accepted |
| | Pair 2 | Pre and scale-based assessment | −9.23 | 0.000* | $\bar{X}_{\text{pre self-assessment}} < \bar{X}_{\text{scale-based assessment}}$ | $H_8$ accepted |
| | Pair 3 | Post and scale-based assessment | −6.92 | 0.000* | $\bar{X}_{\text{post self-assessment}} < \bar{X}_{\text{scale-based assessment}}$ | $H_9$ accepted |

Note:
* The mean differences between sub-groups are statistically significant α = 0.05 level.

on DC. The signs of all significant coefficients were consistent with the research hypotheses, according to the results in Table 4. The R² of the multiple linear regression model was 0.164% and, 16.40% of the variance for the endogenous variable (DC) was explained by exogenous variables (age, DTE, PWE, FTEBC, and PTEBC). In addition, the model guaranteed basic regression assumptions (normality, multicollinearity, heteroscedasticity *etc.*), and the results promoted the statistical validity of the analysis.

To determine any statistically significant differences between pre- and post-assessment means, a paired sample t-test was employed, as shown in Table 5. Results showed that there were statistically significant differences between pre- and post-self-assessment ($p < 0.05$) for the first pair, pre- and scale-based assessment ($p < 0.05$) for the second pair, and post and scale-based assessment ($p < 0.05$) for the third pair.

## DISCUSSION

According to the scale-based DC assessment results (Fig. 4), the DC of most participants was intermediate (B1 (44.1%) and B2 (29.7%)). *Redecker (2017)* defines integrators (B1) as educators who are aware of the potential use of DT but are still experimenting with them in various environments for their own needs. They need consistent practice in their authentic teaching and learning environments to make accurate decisions about DT. For experts

(B2) (*Redecker, 2017*) claims that they have their own DT repertoire ready to be utilized in different situations based on their own experiences and knowledge of DT.

The accumulation of the majority of the DC scores (%73.8) at intermediary levels (B1 or B2) was similar to the other resent observations regarding higher education (*Guillén-Gamez & Mayorga-Fernandez, 2020*; *Jorge-Vazquez et al., 2021*). Considering the high rate of DT usage in their private lives and the abundance of DT facilities in the workplace (Table 3), result of integrator or expert is not a surprise.

Participants responded that the most frequently DT they used were presentations, video and audio, and online/virtual learning environments (Table 2), commonly for information presentation or content-sharing purposes, where the teacher's role was similar to that in traditional face-to-face instruction. Conversely, other tools in Table 2, such as creating video and audio, posters, mind maps, planning tools, online quizzes or polls, and blogs or wikis received relatively low usage. Those tools are commonly used for increasing interaction, communication, and active student participation (*Beldarrain, 2006*). A new faculty development program introducing and highlighting the pedagogical use and benefits of those tools might be beneficial.

Moreover, it was found that the use of sophisticated DTs, like augmented/virtual reality and simulations, came in last and was used at a low level. Although integrating these tools into education requires complex peripherals, a certain level of digital literacy and tailored scenarios, given their valuable pedagogical benefits in higher education compared to other tools, more institutional support is recommended to provide them for faculty members since these technologies are not affordable or easily accessible individually to possess or develop most of the time.

Only 14.4% of participants' DC level achieved a high proficiency status (leader 13.7% and pioneer 0.7%) (Fig. 4). These participants seemed to be competent to lead others and create innovative approaches with DT. Studies with similar results have highlighted that an increase in the DC of faculty members requires not only accessing DT in work environment and daily life, but also in customized, continuous, and convenient personal development opportunities (*Zakharov et al., 2022*; *Lucas et al., 2021*).

A small percentage of participants were in lower (newcomer 1.0% and explorer 10.8%) levels (Fig. 4). These participants may have a low usage of DT in their private life and/or were employed in an inadequate work environment. Further research and comprehensive need analysis for those staff could provide better and tailored faculty development strategies.

Based on the result of the multiple linear regression model (Table 4), all hypotheses, except the second one, were accepted. In the first accepted hypothesis, it was found that age had a negative effect on DC which is in line with many studies (*Cruz & Díaz, 2016*; *Gallardo-Echenique, Poma & Esteve, 2018*; *Gonzalez, Amaro & Martinez, 2019*). Given the high private use ratio of DT (Table 3), younger participants may have higher self-confidence. *Lucas et al. (2021)* claimed that undergraduate or pre-service programs of younger faculty members have more DT courses and a higher likelihood of using DT in training.

By rejecting the second hypothesis of "teaching experience in years" (TE) and accepting the third hypothesis, "using DT in teaching" (DTE), it is possible to claim, like *Guillén-Gamez et al. (2021)*, that DC is more related to using DT for teaching purposes. While some studies claim that lack of TE would be a barrier to proper technology integration (*Batanero et al., 2021*), others argue that TE may be effective in sub dimensions of DC such as "communication and collaboration, digital content creation, security and problem solving" (*Zhao et al., 2021*, p.11).

The "perception of how the work environment meets the requirements for using DT" (PWE) seems to affect DC positively, as *Cattaneo, Antonietti & Rauseo (2022)* have found. In the study, there appeared to be a wide range of online communication, content management and presentation tools, but a limited number of field-specific tools were provided by the participants' institutions. This would also help to understand faculty members' intermediate DC levels. It is difficult to use and gain field-specific DT experience for faculty members if they do not have institutional support. In this respect, institutions should be responsible for providing a proper need-analysis, teacher training and continuous support mechanisms.

Furthermore, even though this study tried to obtain experiences of faculty members in online teaching at pre COVID-19 times, it might not be possible to eliminate the effects of compulsory distance education period (*Shagiakhmetova et al., 2022*). In those teaching efforts, faculty members may have needed to use various synchronous and asynchronous communication software, learning management systems and related ICT tools intensely.

The next two accepted hypotheses ($H_5$ and $H_6$) showed that the experience participants had in teaching in fully- or partially-online experiences (FTEBC and PTEBC, respectively) had a positive effect on DC levels, as DTE had. Based on the assumption that any competence improves with practice (*Cattaneo, Antonietti & Rauseo, 2022*), faculty members with experience in implementing online courses are often expected to be more competent in DT. As online teaching involves intense use of DT (*Erlam et al., 2021*), experience in such a teaching environment may be a sound indicator of the DC of faculty members (*Thalheimer, 2017*).

By accepting the last three hypothesis ($H_7$, $H_8$, and $H_9$ in Table 5), it is possible to claim that the participants underestimated their DC capacity. This result supported the idea that an individual's self-perceived level of knowledge may not reflect their actual level since they were unaware of it (*Alba & Hutchinson, 2000*). Furthermore, *Moore & Cain (2007)* claimed that this fault estimation is more in difficult skill-based tasks. Although the DC levels assigned by DigCompEdu framework (*DigCompEdu, 2021*) were explained during the self-assessment process, it was discovered that inquiring about their actions, attitudes, and roles in using DT provided a more accurate DC assessment.

## Limitations

The study findings and conclusions have certain limitations. The self-reflective nature of the DC assessment *via* "DigCompEdu Checkin" (*DigCompEdu, 2021*), the use of convenient sampling and the cross-sectional design limits generalization of the results. The DC of the faculty members may change over time; therefore, further research with a

longitudinal design should be conducted. Beyond the DigCompEdu self-reflection questionnaire, other aspects of DC may be supplemented by alternative assessment techniques, such as focus group interviews or task/performance-based observations.

## CONCLUSIONS

The study showed that the majority of the faculty members teaching in health sciences have intermediate (B1 or B2) levels of DC. According to the *DigCompEdu (2021)* framework, this intermediate level means they are capable of using several DT in teaching and overcoming their own problems. Others, with small percentages, are at the upper (about 14%) or lower (about 11%) ends of the DC level continuum.

In order to understand what features would have affected the measured DC level, a regression analysis provided influencing factors such as age, digital teaching experience, perception of work environment, and having teaching experience in online/partially online courses were identified as influencing factors for DC in the current study. These findings can be used to develop new strategies for developing the digital competencies of faculty members in higher education.

One recommendation for policy makers is to plan and implement tailored faculty development training programs to enhance DC. The design of training programs should be informed by an understanding of the existing levels of DC among the faculty members, with the aim of providing a diverse range of learning scenarios that cater to the varying needs of faculty members with different DC levels. Rather than adopting a one-size-fits-all approach, training programs should be tailored to the specific requirements of different groups of faculty members. Enhancing DC with supportive strategies for faculty members would increase self-confidence and their autonomy in overcoming obstacles like the outbreaks in the pandemic (*Chang, Gaines & Mosley, 2022*).

Institutions with a significant number of upper-level faculty members have an opportunity to arrange for and encourage the collaborative knowledge, and information and experience exchange activities among academicians with different levels of DC. Peer learning and collaborative learning would motivate faculty members to participate more readily in such personal development training activities (*Langset, Jacobsen & Haugsbakken, 2018*).

Another recommendation is to enrich the learning environment with proper DTs designed for daily needs, like information access and communication, as well as for field-specific and sophisticated pedagogic outcomes. As this study showed, training about DC should be supported with sustainable and continuous facilities for faculty members for successful DT integration in teaching.

### Funding

This research was undertaken as part of the Erasmus+ project, titled "Digital and Hybrid Teaching and Learning of Practical Skills in Higher Education (DITEPRACT)" (Project

No. 2020-421 1-FI01-KA226-HE-092515). The funders had no role in study design, data collection and analysis, decision to publish, or preparation of the manuscript.

## Grant Disclosures
The following grant information was disclosed by the authors:
Erasmus+ Project: 2020-421 1-FI01-KA226-HE-092515.

## Competing Interests
The authors declare that they have no competing interests.

## Author Contributions
- Halil Ersoy conceived and designed the experiments, performed the experiments, prepared figures and/or tables, and approved the final draft.
- Cigdem Baskici conceived and designed the experiments, performed the experiments, analyzed the data, prepared figures and/or tables, and approved the final draft.
- Aydan Aytar conceived and designed the experiments, performed the experiments, authored or reviewed drafts of the article, and approved the final draft.
- Raimonds Strods performed the experiments, authored or reviewed drafts of the article, and approved the final draft.
- Nora Jansone Ratinika performed the experiments, authored or reviewed drafts of the article, and approved the final draft.
- António Manuel Lopes Fernandes conceived and designed the experiments, analyzed the data, authored or reviewed drafts of the article, and approved the final draft.
- Hugo Neves analyzed the data, authored or reviewed drafts of the article, and approved the final draft.
- Aurelija Blaževičienė performed the experiments, analyzed the data, authored or reviewed drafts of the article, and approved the final draft.
- Alina Vaškelytė performed the experiments, authored or reviewed drafts of the article, and approved the final draft.
- Camilla Wikström-Grotell performed the experiments, analyzed the data, authored or reviewed drafts of the article, and approved the final draft.
- Heikki Paakkonen performed the experiments, authored or reviewed drafts of the article, and approved the final draft.
- Anne Söderlund performed the experiments, authored or reviewed drafts of the article, and approved the final draft.
- Johanna Fritz analyzed the data, authored or reviewed drafts of the article, and approved the final draft.
- Sultan Kav conceived and designed the experiments, analyzed the data, authored or reviewed drafts of the article, and approved the final draft.

## Ethics
The following information was supplied relating to ethical approvals (*i.e.*, approving body and any reference numbers):

Ethical approval was granted by the ethics board committee of the corresponding author's university, named Başkent University Academic Evaluation and Assessment Coordina-tion Office on May 26th, 2021, with document number of "E-62310886-604.02.01-35305".

## Data Availability

The raw data is available in the Supplemental File.

## Supplemental Information

Supplemental information for this article can be found online at http://dx.doi.org/10.7717/peerj.18456#supplemental-information.

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
