# Peer review of "Digital competence of faculty members in health sciences measured via self-reflection: current status and contextual aspects"

_PeerJ, doi:10.7717/peerj.18456_

## Round 0.1 · original submission · Major Revisions

The reviewers were very positive regarding your manuscript entitled. They acknowledged the quality and relevance of the work, highlighting its originality and significant contribution to the field of study. However, several aspects have been identified as areas for improvement and should be given due attention to ensure the article reaches its full potential.

Reviewer 1 ·

Basic reporting

No comment

Experimental design

No comment

Validity of the findings

No Comment

Reviewer 2 ·

Basic reporting

This paper is titled, Digital Competence of Faculty Members in Healthcare and Social Science via Self-Reflection, however, the keywords listed do not seem to reflect the title. The language used requires consistency in the verb tenses. In addition, I am under the impression that this paper does not fit the scope of the journal. The conclusion is short and abrupt. might want to remind the readers of the significance of the research and avenues for future research.

Experimental design

The Materials & Methods section was elaborate and detailed which includes ethical considerations. For this section,
"At the beginning of the study, the written permission was obtained via email from the
DigCompEdu research team. The original language of the questionnaire was English,
nevertheless other version in some European languages were also available (DigComEdu, 2021).
Two partners used English version and others preferred a version in their local language. The
Cronbachís ] coefficient of questionnaire was calculated as 0.915 for the study"

Who is the DigCompEdu team? Was this introduced earlier? This section is too wordy and can be rephrased for clarity.

Validity of the findings

The findings are valid, well-explained and elaborate.

Additional comments

While I enjoyed reading your paper, I am still unclear about the rationale behind collecting data from faculty members in both healthcare and social science fields. Was there a motivation to compare these two groups? Why not focus solely on healthcare? What is your ultimate goal? Were you hoping that the data collected would help enhance their digital competence skills through professional development? This could be further elaborated in the text.

Reviewer 3 ·

Basic reporting

Strengths:
• The manuscript is clear and well-structured, providing a comprehensive review of literature and a detailed explanation of digital competence.
• It effectively uses tables and figures to present data, enhancing the clarity of results.

Areas for Improvement:
• The manuscript could improve by tightening the language to eliminate minor grammatical errors and enhance readability for an international audience.

Experimental design

Strengths:
• The research design is methodologically sound, using established frameworks (DigCompEdu) to assess digital competence.
• The study covers a significant sample size, enhancing the robustness of its conclusions.

Areas for Improvement:
• The methods section could be expanded to more clearly explain the process of data analysis, particularly how the responses were quantitatively and qualitatively evaluated.

Validity of the findings

Strengths:
• Findings are supported by clear statistical evidence, indicating robust analysis.
• Conclusions are well-aligned with the research questions and are logically derived from the data.

Areas for Improvement:
• The discussion could delve deeper into how these findings could influence policy changes in educational institutions regarding digital competence training.

·

Basic reporting

The text of the article is clear and uses a good level of English, which facilitates readers' understanding and ensures the professionalism of the work. The references used are adequate and pertinent to the topic, providing a solid theoretical foundation. However, the article could include a more in-depth discussion on professor autonomy and the choice of professors in the teaching and learning processes, which would further enrich the content and promote broader reflection on these crucial aspects.

The structure of the article, as well as the presented graphs and tables, are adequate and contribute to the clarity and organization of the information. These visual elements aid in data comprehension and make the article more dynamic and accessible. The formatting and visual presentation comply with academic standards, ensuring the professionalism of the work and facilitating critical reading by evaluators and other readers.

The results presented are relevant and offer significant contributions to the hypotheses raised in the study. However, the analysis of the Covid-19 pandemic could be conducted more critically. The authors dedicated only one paragraph to the study results related to the pandemic, despite all the hypotheses in the work being, in some way, affected by this global event. Considering that the study covers the years 2020 and 2021, a more critical analysis of the pandemic's impact would be beneficial for a more complete and in-depth understanding of the results obtained.

Experimental design

This is an original study within the scope of the journal. The study addresses two research questions. The first question is on line 119, asking about the level of digital competencies among faculty members. This research question is satisfactorily answered by the study.

The second question is on line 120, asking about the specific factors that may affect the digital competencies of faculty members. This question is not adequately answered, as the study's hypotheses do not allow the faculty members to respond satisfactorily beyond the options imposed by the authors. Therefore, I believe the second question should be excluded.

The study only addresses the first research question. The research fills a knowledge gap regarding the digital competencies of faculty members during the pandemic period, although the authors refuse to acknowledge this.

The study is conducted with a high technical and ethical standard, and the methods are described in sufficient detail for replication in other pandemic periods.

Validity of the findings

The study is innovative, although it is not as impactful as it could be if it brought a more critical view of the effects of the pandemic period on the school environment. The lack of an in-depth analysis of this context limits the full understanding of the changes and challenges faced by the school community during the pandemic.

I reaffirm that the article could include a more in-depth discussion on professor autonomy and the choice of professors in the teaching and learning processes. Although the literature on digital competencies is sufficient, including these discussions would enrich the debate and provide a more comprehensive perspective on the competencies and challenges faced by professors.

The underlying data presented in the study are adequate and provide a solid basis for the conclusions. However, the data analysis could be expanded to better explore the nuances and variables that affect professors' digital competencies, providing a more detailed and critical view of the results.

The study's conclusion should be rewritten to present the most important evidence about the level of digital competencies of the studied faculty members. A more robust and detailed conclusion would strengthen the study's impact and more clearly demonstrate how the findings contribute to the field of study, as well as highlight practical implications for the training and development of professors.

---

## Round 0.2 · accepted · Accept

Thank you for submitting the revised version of your manuscript. After carefully reviewing the changes, I can confirm that all reviewers' comments and suggestions have been appropriately addressed.

Based on this assessment, I am pleased to inform you that the manuscript is now ready for publication. Congratulations.

Reviewer 3 ·

Basic reporting

The authors have addressed the previous comments and made the necessary changes, the manuscript appears to be ready for acceptance.

Experimental design

The authors have addressed the previous comments and made the necessary changes, the manuscript appears to be ready for acceptance.

Validity of the findings

The authors have addressed the previous comments and made the necessary changes, the manuscript appears to be ready for acceptance.

·

Basic reporting

No comment.

Experimental design

No comment.

Validity of the findings

No comment.